# The Potential of Probiotics to Eradicate Gut Carriage of Pathogenic or Antimicrobial-Resistant *Enterobacterales*

**DOI:** 10.3390/antibiotics10091086

**Published:** 2021-09-08

**Authors:** Yuan-Pin Hung, Ching-Chi Lee, Jen-Chieh Lee, Pei-Jane Tsai, Po-Ren Hsueh, Wen-Chien Ko

**Affiliations:** 1Department of Internal Medicine, Tainan Hospital, Ministry of Health and Welfare, Tainan 700, Taiwan; yuebin16@yahoo.com.tw; 2Department of Internal Medicine, College of Medicine, National Cheng Kung University Hospital, National Cheng Kung University, Tainan 704, Taiwan; chichingbm85@yahoo.com.tw (C.-C.L.); jclee.eric@msa.hinet.net (J.-C.L.); 3Clinical Medicine Research Center, College of Medicine, National Cheng Kung University Hospital, National Cheng Kung University, Tainan 704, Taiwan; 4Department of Medical Laboratory Science and Biotechnology, College of Medicine, National Cheng Kung University, Tainan 705, Taiwan; peijtsai@mail.ncku.edu.tw; 5Institute of Basic Medical Sciences, College of Medicine, National Cheng Kung University, Tainan 705, Taiwan; 6Department of Pathology, National Cheng Kung University Hospital, National Cheng Kung University, Tainan 704, Taiwan; 7Departments of Laboratory Medicine and Internal Medicine, China Medical University Hospital, School of Medicine, China Medical University, Taichung 404, Taiwan; 8Department of Medicine, College of Medicine, National Cheng Kung University, Tainan 705, Taiwan

**Keywords:** probiotics, synbiotics, antimicrobial-resistant, *Enterobacterales*, gastrointestinal tract, livestock

## Abstract

Probiotic supplements have been used to decrease the gut carriage of antimicrobial-resistant *Enterobacterales* through changes in the microbiota and metabolomes, nutrition competition, and the secretion of antimicrobial proteins. Many probiotics have shown *Enterobacterales*-inhibiting effects ex vivo and in vivo. In livestock, probiotics have been widely used to eradicate colon or environmental antimicrobial-resistant *Enterobacterales* colonization with promising efficacy for many years by oral supplementation, in ovo use, or as environmental disinfectants. In humans, probiotics have been used as oral supplements for infants to decease potential gut pathogenic *Enterobacterales*, and probiotic mixtures, especially, have exhibited positive results. In contrast to the beneficial effects in infants, for adults, probiotic supplements might decrease potentially pathogenic *Enterobacterales*, but they fail to completely eradicate them in the gut. However, there are several ways to improve the effects of probiotics, including the discovery of probiotics with gut-protection ability and antimicrobial effects, the modification of delivery methods, and the discovery of engineered probiotics. The search for multifunctional probiotics and synbiotics could render the eradication of “bad” *Enterobacterales* in the human gut via probiotic administration achievable in the future.

## 1. Introduction

Trillions of bacteria colonize in various anatomical locations in the human body, including the mouth, the upper airways, the skin, the vagina, the genitourinary system, and the intestinal tract. These colonized locations represent a highly integrated ecosystem collectively called “microbiota” [1,2]. Thus, humans are considered to be metaorganisms (also termed superorganisms or holobionts) [1,2]. The overlap of the phylogenetic trees of bacterial microbiota and primates suggests the coevolution, especially the genetic coevolution, between host and microbiota [2,3,4]. The microbial colonization of the human body starts immediately following birth, and the community composition is shaped by various environmental factors [5]. The infant gut microbiota is mostly predominated by the members of *Actinobacteria, Proteobacteria, Firmicutes*, and *Bacteroidetes* [5]. Factors influencing microbiome composition and diversity include the mode of delivery, the feeding type, maternal antibiotic and probiotic use, dietary intake, pre-pregnancy body mass index, gestational weight gain, diabetes mellitus, mood, and others [6]. For example, vaginally delivered (SVD) and breast-fed (BF) infants had a higher abundance of gut microbiota than caesarean-section-delivered, milk-powder-fed, and mixed-fed infants [7]. The genera *Enterobacterales* and *Bifidobacterium* were highly abundant in the SVD and BF groups [7]. Prior antibiotic therapy was independently associated with the carriage of extended-spectrum β-lactamase (ESBL)-producing *Enterobacterales* in an infant cohort upon admission to a tertiary teaching hospital in France [8]. Moreover, neonatal enteral tube feeding has been noted to serve as loci for colonization by the members of *Enterobacterales* [9]. Although established during infancy, the complex gut microbial community will be shaped by further medical interventions and societal preferences, such as caesarean section, formula feeding, and antibiotic use [10].

The microbiota in the gut of patients with diseases or who are aging, compared to the relative healthy population, is characterized by a decrease in diversity, greater interindividual variability, fewer beneficial microbes, such as the *Firmicutes*, *Bifidobacterium*, and *Clostridium* species and *Faecalibacterium*
*prausnitzii*, and more pathogenic *Enterobacterales* [11]. The carriage of *Enterobacterales* in the gut is associated with lower phylogenetic diversity, dysbiotic microbiota, and the depletion of anaerobic commensals in the gut microbiota [12,13]. Moreover, among persons with gut colonization by carbapenem-resistant *Enterobacterales* (CRE), compositional and functional changes in the microbiota are linked to an increased risk in subsequent systemic infection and bacteremia [12].

The prevalence rate of antimicrobial-resistant organisms (AMROs), including ESBL-producing *Escherichia coli* and CRE, has increased in recent years [14]. Fortunately, these AMROs have been suppressed by the supernatant of some probiotics, such as *Clostridium butyricum, Enterococcus faecium,* and *Lactobacillus plantarum,* in a dose-dependent manner ex vivo [15]. Thus, it has been suggested that oral probiotic supplements can be used to eradicate *Enterobacterales* colonization in the gut.

Oral antibiotics, probiotics, and fecal microbiota transplantation have recently been analyzed for their potential use in decolonizing ESBL-producing *Enterobacterales* or CRE in the gut over the past 10 years [16]. However, in a review in 2019, Gaud Catho et al. suggest that there is not enough available evidence to recommend these decolonization strategies for the intestinal carriage of antimicrobial-resistant *Enterobacterales* in routine clinical practice [16]. Although the results of the routine clinical practice of probiotics in eradicating the gut carriage of antimicrobial-resistant *Enterobacterales* were inconclusive before 2019, many subsequent ex vivo, in vivo, and animal studies are now ongoing [17,18,19,20,21,22,23].

### 1.1. Rationale for Probiotic Supplements to Eradicate Enterobacterales Carriage in the Gut

Probiotics, by definition, are live microorganisms, and should remain viable when they reach the intended site of action, which is typically the cecum and/or the colon [24]. Most probiotics originate from fermenting food, an ancient form of preservation ingrained in human societies around the world [25]. The microbiome of all fermented foods shows increasing amounts of *Lactobacillales* during the fermentation process, which replaces the initial dominant composition of *Enterobacterales* in these foods [25]. The incorporation of probiotics into food results in higher counts of lactic acid-producing bacteria and lower counts of *Enterobacterales* [26]. To date, probiotics have been widely used as food additives.

The eradication of pathogenic *Enterobacterales* by supplementation with probiotics has been confirmed in several animal models [17,18,19,20,23]. Mice pretreated with *B. bifidum* ATCC 29521 exhibited a significant increase in the diversity of the gut microbiome, and a decrease in the abundance of the genus *Escherichia-Shigella*, belonging to the family *Enterobacterales* [17]. These changes in microbiota after *B. bifidum* ATCC 29521 pretreatment were associated with a decrease in the severity of inflammatory bowel disease [17]. Moreover, *L. rhamnosus* GG could reduce the mortality rate of septic mice by modulating gut microbiota composition, especially reducing the lipopolysaccharide producers, such as *Enterobacterales* [18]. *Bacillus coagulans* SANK 70258 suppressed *Enterobacterales* and enhanced butyrogenesis in microbiota models [19]. *L. plantarum*, isolated and identified from yak yogurt, increased the content of beneficial bacteria, including *Bacteroides, Bifidobacterium*, and *Lactobacillus,* and reduced the content of harmful bacteria, including *Firmicutes, Actinobacteria, Proteobacteria*, and *Enterobacterales*, and, thus, could protect against alcoholic liver injury [20]. The oral administration of *L. rhamnosus* GG can improve the survival rate of mice with sepsis by reducing lipopolysaccharide-producing *Enterobacterales*, decreasing epithelial apoptosis, and increasing the proliferation of colonic epithelium and the expression of tight junction proteins [23]. A mixture of probiotics showed more efficient eradication of pathogenic *Enterobacterales* in vivo. In mice, the mixture of *L. fermentum* GOS47 and *L. fermentum* GOS1 significantly decreased the viable count of *Enterobacterales* with potential anti-inflammatory activity and short-chain fatty acid production [27]. Thus, the favorable effect of probiotic supplements on at least the partial elimination of pathogenic *Enterobacterales*, ex vivo and in vivo, has promoted their application in clinical diseases.

Supplementation with probiotics has been investigated for the alleviation of the disease severity of systemic or gastrointestinal inflammatory diseases, such as sepsis, inflammatory bowel disease, and chemotherapy- or radiation-induced gastrointestinal mucositis [17,18,19,28,29]. For example, patients receiving cytotoxic and radiation therapy showed striking alterations in intestinal microbiota with, most frequently, a decrease in *Bifidobacterium, Clostridium* cluster XIVa and *F. prausnitzii,* and an increase in *Enterobacterales* and *Bacteroides* [28]. These pathogenic alterations resulted in the development of mucositis and bacteremia [28,29]. The prevention of cytotoxic chemotherapy-induced mucositis by probiotics has been investigated in randomized clinical trials with some promising results. Moreover, in a meta-analysis of randomized controlled trials with patients undergoing a colorectal resection, the perioperative administration of probiotics or synbiotics was associated with increased numbers of *Lactobacillus* and decreased counts of *Enterobacterales* [30]. These changes in gut microbiota were associated with less diarrhea, less symptomatic intestinal obstruction, and a lower incidence of total postoperative infections [30]. Accordingly, the use of probiotics in modulating gut microbiota and decreasing pathogenic *Enterobacterales* has become popular for application in many bowel or extra-bowel diseases, and more extensive probiotic usage can be expected in the future.

### 1.2. Probiotic Supplements to Decrease Gut Carriage of Enterobacterales in Livestock or Domesticated Animals

The use of probiotics in preventing gut *Enterobacterales* colonization has been applied in livestock breeding [31,32,33,34,35,36]. *Lactobacillus* supplementation, in directly fed microbes or used as phytobiotic feed additives, reduced the prevalence of ESBL-producing *Enterobacterales* in broilers [31]. In young broilers, the neonatal colonization of *Enterobacterales* strains led to immune dysregulation and chronic inflammation, but early life exposure to a mixture of probiotics containing lactic-acid-producing bacteria could modulate the immune functions through the activation and trafficking of immune cells [32]. In weaned piglets, *B. subtilis* DSM25841 treatment reduced enterotoxigenic *E. coli* (ETEC) F4 infection and decreased the risk of diarrhea [34]. *L. reuteri* KUB-AC5 possessed antimicrobial activity in reducing *Salmonella* contamination in live poultry [35]. The above data further support the use of probiotics as feed additives in livestock breeding.

Other than oral intake, the in ovo administration of probiotics for eradicating gut *Enterobacterales* colonization has been used in chickens [37,38]. Via the in ovo route during hatching, a *Bacillus*-based probiotic (BPP) can reduce the severity of the virulent *E. coli* horizontal transmission among broiler chickens, which might be achieved by alterations in the microbiota composition, including a decrease in *Enterobacterales* and an increase in *Lachnospiraceae* [37]. In another chicken study, the in ovo administration of lactic-acid-producing bacteria resulted in an increased abundance in the *Lactobacillaceae* family and *Lactobacillus* genus, and a decrease in *Enterobacterales* and *Enterococcaceae* [38]. For bird species, the early in ovo administration of probiotics seems to be more efficient in eradicating gut *Enterobacterales* colonization before hatching.

A mixture of probiotics may work better to eradicate gut *Enterobacterales* in livestock breeding [33,39,40,41]. The administration of multistrain probiotics containing *L. acidophilus* LAP5, *L. fermentum* P2, *Pediococcus acidilactici* LS, and *L. casei* L21 could modulate intestinal microbiota (increase *Lactobacillaceae* abundance and reduce *Enterobacterales* abundance), increase the gene expression of tight junction proteins (ZO-1 and Mucin 2) and the immunomodulatory activity (downregulation of mRNA levels of interferon-γ [IFN-γ] and lipopolysaccharide-induced tumor necrosis factor-α [TNF-α], and upregulation of IL-10) in broiler chickens [33]. Commercially available synbiotics, either BioPlus 2B^®^ or Cylactin^®^ LBC, had a more significant impact on the concentration of lactic acid, short-chain fatty acids (SCFAs), and branched-chain fatty acids (BCFAs), than a single probiotic in sows [39]. Mixed probiotics composed of three thermophilic lactic-acid-producing bacteria (LAB) strains, *L. helveticus* BGRA43 (strong proteolytic activity, antimicrobial activity, and adhesion to gut cell activity), *L. fermentum* BGHI14 (immunomodulatory effect), and *Streptococcus thermophiles* BGVLJ1–44 (strong proteolytic activity and immunomodulatory effect), influenced the colonization of piglet guts with beneficial bacteria, and reduced the number of *Enterobacterales* in some treated sows [41]. Thus, the commercially available mixed regimens of probiotics may be more efficient in eliminating *Enterobacterales* carriage in the guts of livestock.

Furthermore, probiotics in combination with prebiotics (foods that promote the growth of beneficial microbes), or phytobiotics (plant-derived products), have been utilized in livestock breeding for the eradication of gut colonization by *Enterobacterales* [40,42]. *Lactobacillus* strains (*L. agilis* and *L. salivarius*), combined with phytobiotics, have been used to reduce the survival of potentially problematic bacteria, such as ESBL-producing *E. coli* in broilers [42]. The synbiotics (*L. rhamnosus* HN001 and *P. acidilactici*) combined with the phytobiotics (*Agave tequilana* fructans) induced morphological modifications in the duodenal mucosa of broilers that, in turn, promoted resistance to infections caused by *S. typhimurium* and *C. perfringens* [40].

In addition, a probiotic-based cleaning strategy to decontaminate *Enterobacterales* in livestock environments has been reported [43]. The cleaning product, containing *B. subtilis, B. pumilus*, and *B. megaterium* spores, was used to clean fresh and reused broiler litters [43]. These *Bacillus* spores were able to successfully colonize reused poultry litters to decrease the mean counts of total aerobic bacteria, *Enterobacterales*, and coagulase-positive *Staphylococcus* [43]. A decrease in *Enterobacterales*, mainly the genus *Escherichia*, was also observed in the ceca of broilers reared on reused litters treated with the cleaning product [43]. The efficacy and safety issues of this probiotic-based cleaning product are still ongoing for livestock environments, but have not been tested for human environments.

Among domesticated animals, such as weaning rabbits, *L. buchneri* could decrease *Enterobacterales* counts in the gut and upregulate anti-inflammatory interleukin (IL)-4 and the expression of intestinal barrier-related genes, such as zonula occludens-1 (ZO-1), and, thus, may prevent diarrhea [36]. In a randomized controlled trial of healthy cats, *Enterobacterales* declined after the administration of synbiotics, a combination of probiotics (Proviable-DC^®^ containing *E. faecium, B. bifidum, E. thermophilus, L. acidophilus, L. bulgaricus, L. casei,* and *L. plantarum*) [21]. Among dogs fed Queso Blanco cheese with *B. longum* KACC 91563 for eight weeks, a reduction in harmful bacteria, such as the *Enterobacterales* and *Clostridium* species, was noted [22]. The successful decrease in *Enterobacterales* after probiotic supplementation in pet animals arouses hope for the eradication of gut *Enterobacterales* carriage via the use of probiotics in humans.

### 1.3. The Selection of Probiotics to Decrease Gut Colonization of Enterobacterales in Humans

The common, safe, and well-studied probiotics used to eradicate the gut carriage of *Enterobacterales* in humans include the *Lactobacillus* [44,45,46,47] and *Bifidobacterium* [17,47,48] species. In extremely low-birth-weight infants, *L. reuteri* supplementation for one week resulted in a lower abundance of *Enterobacterales* and *Staphylococcaceae* [44]. Among infants fed *B. infantis* EVC001, a high abundance of *Bifidobacteriaceae* developed rapidly with a reduced abundance of antibiotic-resistant genes among *Enterobacterales* and/or *Staphylococcaceae* [48].

As noted in livestock, probiotic mixtures might provide better protection against gut *Enterobacterales* colonization than a single probiotic regimen in humans [45,49,50,51]. A probiotic mixture (Bactiol duo^®^) containing *Saccharomyces boulardii*, *L. acidophilus* NCFM, *L. paracasei* Lpc-37, *B. lactis* Bl-04, and *B. lactis* Bi-07, provides better eradication of AmpC-producing *Enterobacterales* carriage than *S. boulardii* CNCM I-745^®^ [45]. Oral daily supplementation with a combination of a prebiotic (Emportal^®^: lactitol) and probiotics (Infloran^®^: *B. bifidum* and *L. acidophilus*) for three weeks decreased the intestinal load of OXA-48-producing *Enterobacterales* among eight patients with long-term intestinal carriage [49]. Moreover, the ingestion of combined probiotics containing *L. plantarum* LK006, *B. longum* LK014, and *B. bifidum* LK012 could significantly reduce the abundance of *Enterobacterales* and increase the abundance of *Lactobacillaceae* in preterm infants [50]. These changes in microbiota were correlated with a decreased serum inflammatory cytokine level of IL-6 and improved the survival rate of these infants. A mixture of *B. breve* M-16V, *B. longum* subsp. *infantis* (*B. infantis*) M-63, and *B. longum* subsp. *longum* BB536, achieved significantly higher levels of *Bifidobacterium*-predominant microbiota and lower detection rates for *Clostridium* and *Enterobacterales* than a single *B. breve* strain [51]. For human safety, the most common probiotics for combination are the *Lactobacillus* and *Bifidobacterium* species.

### 1.4. Probiotic Supplementation to Decrease Potential Gut Pathogenic Enterobacterales from Infants to Children

Probiotics have been used as supplements for infants to decease potential gut pathogenic *Enterobacterales* [44,48,50,52,53,54,55,56,57] (Table 1). Among hospitalized infants, early administration of *L. reuteri* DSM 17938 was associated with less colonization by diarrheagenic *E. coli* [55]. In a randomized placebo-controlled study that administered *B. infantis* to 24 infants with gastroschisis, the microbial communities were not significantly influenced [52]. In a double-blind, placebo-controlled randomized clinical study conducted on 69 preterm infants, *B. lactis* BB-12 supplementation resulted in lower viable counts of *Enterobacterales* [57]. Moreover, in a randomized trial of 300 healthy newborns, the receipt of *B. longum* BB536 was associated with a higher *Bifidobacterium/Enterobacterales* ratio (B/E), an increased number of IFN-γ-secreting cells, and a higher ratio of IFN-γ/IL-4-secreting cells, which is indicative of the increased Th1 response [54]. Among 21 neonates that underwent surgery for congenital heart disease >7 days after birth, the enteral *B. breve* strain Yakult (BBG-01) supply led to significantly fewer *Enterobacterales* in the gut [56]. Since infants, especially preterm infants, are susceptible to intestinal infection, many probiotic studies have been conducted on these susceptible hosts that have provided promising results against pathogenic *Enterobacterales* colonization in the gut.

However, not all studies have shown the presence of the beneficial effects of the addition of probiotics for infants. In a double-blind randomized control trial, 21 bottle-fed preterm infants receiving *L. rhamnosus* GG did not show a decrease in the numbers of *Enterococcus* and *Enterobacterales* in the gut, increased weight gain, or a decreased hospital stay compared to 26 control infants [58]. In an early review of randomized controlled trials including preterm infants, the *B. animalis* subsp. *lactis* supplement could increase fecal *Bifidobacterium* counts and reduce *Enterobacterales* and *Clostridium* counts, but it did not influence the risk of necrotizing enterocolitis or sepsis [59]. The diverse inhibitory potential of *Enterobacterales,* and the microbiota-modulating effect of probiotics, are likely due to the intrinsic diversity of the gut microbiota of infants and children inhabiting different areas [60].

### 1.5. Probiotic Supplementation to Decrease Gut Pathogenic or Antimicrobial-Resistant Enterobacterales Colonization in Adults

Among adults, probiotic supplements have been shown to decrease, but have failed to totally eradicate, potential antimicrobial-resistant or pathogenic *Enterobacterales* in the gut [45,47,61,62,63,64,65,66] (Table 2). To eradicate potential antimicrobial-resistant *Enterobacterales*, a clinical trial of a probiotic mixture (Bactiol duo^®^: *S. boulardii, L. acidophilus* NCFM, *L. paracasei* Lpc-37, *B. lactis* Bl-04, and *B. lactis* Bi-07) showed that colonization with AmpC-producing *Enterobacterales* transiently increased after amoxicillin-clavulanate therapy and declined after probiotic intervention [45]. To eradicate potential pathogenic *Enterobacterales* in human-immunodeficiency-virus-infected individuals, *L. rhamnosus* GG supplementation was used and resulted in a decrease in intestinal inflammation, along with a reduction in *Enterobacterales* in the gut [62]. The consecutive intake of fermented soymilk (containing isoflavone) and *L. casei* Shirota among 60 healthy premenopausal Japanese women was able to decrease the fecal levels of *Enterobacterales* and to increase isoflavone bioavailability [63].

In contrast to the promising results of the probiotic trials on the eradication of potential antimicrobial-resistant *Enterobacterales* mentioned above, a randomized single-blind, placebo-controlled trial in southern Sweden used a probiotic mixture of eight living bacterial strains, Vivomixx^®^, but the successful eradication of fecal ESBL-producing *Enterobacterales* carriage was rarely observed [47]. Among 31 Danish adults who traveled to India for 10–28 days, the ingestion of *L. rhamnosus* GG had no effect on the risk of ESBL-producing *Enterobacterales* colonization [61]. Of note, in 75 patients who underwent elective colon surgery, the oral intake of *L. plantarum* 299v for one week resulted in increased *Enterobacterales* and Gram-negative anaerobes in the colon, but no change in the incidence of bacterial translocation or postoperative complications [66]. The diverse effect of probiotic supplements on gut *Enterobacterales* carriage is likely due to the different baseline gut microbiota and the decolonization efficacy of a variety of probiotic components. To date, probiotic supplementation is not routinely recommended to replace routine antibiotic decontamination in the preoperative preparation of the digestive tract [67]. However, probiotics or synbiotics might be used in combination with a conventional bowel preparation to reduce the fecal carriage of *Enterobacterales* [68]. However, the majority of larger-scale clinical trials show no evident clinical benefits, such as lower inflammatory responses, fewer infectious complications, or higher survival rates, among adults who consume probiotic supplements.

### 1.6. Possible Mechanisms by Which Probiotic Supplementation Decreases Gut Enterobacterales Carriage

The decrease in gut *Enterobacterales* carriage after probiotic supplementation might be related to the increase in beneficial bacteria, such as *Bacteroides, Bifidobacterium*, and *Lactobacillus* [20,48,50], and SCFA-producing bacteria (such as *Lachnospiraceae* and *Ruminococcaceae*) [69], as well as to the changes in the metabolome in the gut (Figure 1) [39,70]. The continuous intake of a combination of probiotic cheese enriched with *L. reuteri* CCM 8617RIF and crushed flaxseed resulted in an alleviation of the infection course induced by pathogenic *E. coli* O149:F4NAL, favored n-3 polyunsaturated fatty acid metabolism, and inhibited n-6 PUFA metabolism in the gut [70]. Moreover, a probiotic mixture resulted in a greater increase in lactic acid, SCFAs, and branched-chain fatty acids (BCFAs), than a single probiotic in sows [39]. Conclusively, the gut metabolome changes after probiotic supplementation builds up an environment that is not friendly to *Enterobacterales*.

In addition, a microbe is regarded as a beneficial probiotic if it can secrete antimicrobial proteins targeting pathogenic *Enterobacterales*. For example, a probiotic strain, *E. coli* Nissle 1917, secretes small proteins called microcins that possess antimicrobial activity against pathogenic *Enterobacterales* during intestinal inflammation [71]. Therefore, probiotics capable of producing antimicrobial proteins might provide better *Enterobacterales* eradication efficacy. The *Enterobacterales* eradication capacity of probiotics can also result from the nutrition competition between probiotics and *Enterobacterales.* Commensal microbiota contributes to colonization resistance by competing with *Salmonella enteritidis* for oxygen, a resource critical for pathogen expansion [72]. An analysis of the complex nutrition competition in the microbiota of the gut provides an alternative method for selecting appropriate probiotics against *Enterobacterales*.

### 1.7. Improve the Effect of Probiotics in Eradicating Enterobacterales

Changes in the delivery method of probiotics might provide alternative ways of eradicate *Enterobacterales* in the gut [73]. Among patients with mild left-sided ulcerative colitis, the oral intake of *L. casei* DG failed to affect colonic flora, but the rectal administration of the same probiotics increased *Lactobacillus* spp. and reduced *Enterobacterales*, significantly decreased Toll-like receptor (TLR)-4 and IL-1β mRNA levels, and increased mucosal IL-10 [73]. For probiotics vulnerable to gastric acid or intestinal enzymes, rectal administration might provide better efficacy for eradicating *Enterobacterales* in the gut.

Other than naturally found probiotics, engineered probiotics specifically targeting *Enterobacterales* have been investigated to improve gut colonization eradication. The introduction of genes with beneficial effects into probiotics provides additional effects, such as acid resistance, immune modulation, and gut barrier protection effects. A genetically engineered plasmid was delivered to *E. coli* that gained the capacity to produce tetrathionate which can inhibit the growth of *Salmonella* [74].

To facilitate the buildup of healthy gut microbiota, the ex vivo selection of appropriate probiotics is mandatory. Bacteriocin-producing bacteria capable of inhibiting bovine and wastewater *E. coli* isolates have been tested for their activity against Shiga toxin-producing *E. coli*, antimicrobial-resistant *E. coli*, and related enteric pathogens [75]. The selected bacteriocin-producing bacteria show potential as next-generation control strategies in livestock and humans. Another selected probiotic is *B. infantis*, a unique gut bacterium with a prodigious capacity to digest human milk oligosaccharides, that was specifically selected for the focused manipulation of infant intestinal microbiota [76].

### 1.8. Clinical Trials of Probiotics or Synbiotics to Improve Gut Health

There are three ongoing randomized clinical trials on dietary supplementation with probiotics aimed at gut *Enterobacterales* eradication registered at ClinicalTrials.gov, posted from July 2008 to July 2021 (Table 3) [77]. Commercial probiotic mixtures, synbiotics, are being applied in these three trials, and the commonly used probiotic strains include the *Lactobacillus*, *Bifidobacterium,* and *Streptococcus* species. Of note, one interesting trial is comparing the effects of gut *Enterobacterales* eradication between synbiotics and fecal microbiota transplantation.

### 1.9. Clinical Safety Issue of Probiotics

Although the efficacy of probiotics in decreasing the gut carriage of *Enterobacterales* has been recognized, there are still concerns regarding their clinical safety, including potential infections or the inflammatory/fatal effects derived from toxins produced either by the probiotic strains or bacterial contaminants [78]. *Lactobacillus* infections after taking probiotic products containing the *Lactobacillus* species have been reported in immunocompromised patients [79,80,81]. *Lactobacillus* endocarditis has been reported in an otherwise healthy patient taking a probiotic formulation containing *Lactobacillus* [82]. Moreover, there are substantial concerns about the transfer of resistance genes among probiotics, pathogens, and gut microbiota through horizontal gene transfer and the adverse potential of probiotics as the source of antimicrobial resistance genes [83,84]. Thus, the clinical application of probiotics for decreasing *Enterobacterales* gut carriage among patients with an immunocompromised status should consider the possibility of opportunistic infections caused by these probiotic strains.

## 2. Conclusions

The rationale for using probiotic supplements to eradicate gut *Enterobacterales* carriage includes changes in the microbiota and metabolomes, nutrition competition, and the secretion of antimicrobial proteins to establish a gut environment that is not friendly to *Enterobacterales*. Many probiotics indeed do show *Enterobacterales*-inhibiting effects ex vivo and in vivo. In livestock, probiotics have been widely used to eradicate colonic or environmental *Enterobacterales* colonization for years, either administered by oral supplementation, in ovo use, or used as environmental disinfectants. For humans, probiotics have been used as dietary supplements for infants to decease potentially pathogenic *Enterobacterales* in the gut, and probiotics mixtures have shown promising results. This encouraging effect of probiotics on decreasing the gut carriage of *Enterobacterales* is likely related to the simple gut microbiota in infants, and less interference from underlying chronic diseases and prior antimicrobial exposure.

In contrast to the beneficial effects in infants, for adults, probiotic supplements might decrease potentially pathogenic *Enterobacterales* in the gut, but they fail to eradicate them. More efforts to confront dysbiosis resulting from comorbidities or antimicrobial therapy, and to select multifunctional probiotics or synbiotics to improve gut health in elderly patients with complex health problems, are currently required.

## Figures and Tables

**Figure 1 antibiotics-10-01086-f001:**
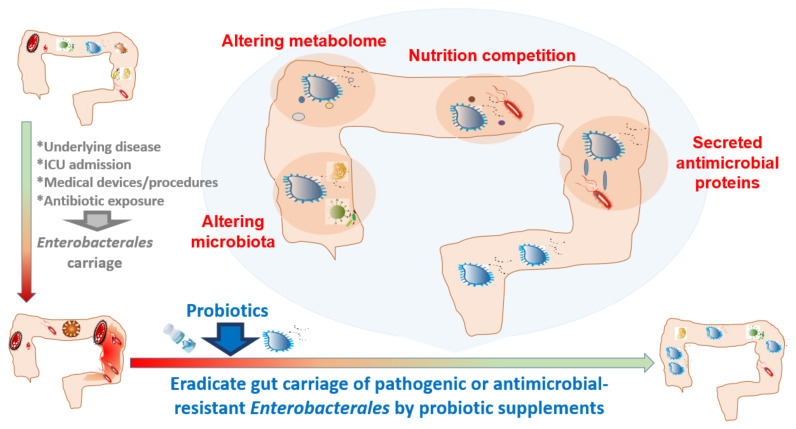
In humans, some clinical settings or interventions might promote intestinal carriage of *Enterobacterales*. Probiotics might be beneficial in eradicating gut carriage of pathogenic or antimicrobial-resistant *Enterobacterales*.

**Table 1 antibiotics-10-01086-t001:** Probiotic supplements for infants and children to decease potential pathogenic *Enterobacterales* in the gut.

First Author	Country	Publish Year	Patient Population/Number	Probiotics	Main Findings after Probiotic Supplementation	References
Mohan R	Germany	2006	Preterm infants/69	*Bifidobacterium lactis* Bb12	Lower viable counts of *Enterobacterales*	[57]
Chrzanowska-Liszewska D	Poland	2012	Bottle fed preterm/60	*Lactobacillus rhamnosus* GG (LGG)	Increase number of *Enterobacterales* in gut	[58]
Umenai T	Japan	2014	Neonates undergoing cardiac surgery/21	*B. breve* strain Yakult (BBG-01)	Significantly fewer *Enterobacterales* in gut	[56]
Savino F	Italy	2015	Hospitalized infant/60	*L. reuteri* DSM 17938	Less colonization by diarrheagenic *E. coli*.	[55]
Wang C	Japan	2015	In preschool and school-age children/23	*L. casei* strain Shirota	Increased population levels of *Bifidobacterium* and total *Lactobacillus*, decreased *Enterobacterales, Staphylococcus* and *Clostridium perfringens*	[53]
Wu BB	China	2016	Healthy newborns/300	*B. longum* BB536	Higher *Bifidobacterium/Enterobacterales* ratio and increased the ratio of IFN-γ/IL-4 secretion cells	[54]
Powell WT	USA	2016	Infants/24	*B. longum* subsp. *infantis*	Overall, microbial communities were not significantly influenced, with trends only toward lower *Enterobacterales*	[52]
Li YF	China	2019	Low birth weight infants/36	*L. plantarum* LK006, *B. longum* LK014, and *B. bifidum* LK012	Increase in *Streptococcaceae* and *Lactobacillaceae* and decrease in *Enterobacterales*	[50]
Nguyen M	USA	2021	Infants/77	*B. infantis*	Reduced abundance of antibiotic resistance genes among *Enterobacterales* and *Staphylococcaceae*	[48]
Martí M	Sweden	2021	First month/132	*L. reuteri*	Lower abundance of *Enterobacterales* and *Staphylococcaceae*	[44]

**Table 2 antibiotics-10-01086-t002:** Probiotic supplements for adults to decease potential pathogenic *Enterobacterales* in gut.

First Author	Country	Publish Year	Patient Number	Probiotics	Main Findings after Probiotic Supplementation	References
Mangell P	Sweden	2012	75	*Lactobacillus plantarum* 299v	Increased *Enterobacterales* and Gram-negative anaerobes in the colon 1 week after probiotics without change in the incidence of bacterial translocation and postoperative complications	[66]
Larsen N	Denmark	2013	50	*L. salivarius* Ls-33	No significant influence on *Clostridium* cluster I, *Clostridium* cluster IV, *Faecalibacterium prausnitzii, Enterobacterales, Enterococcus,* the *Lactobacillus* group, and *Bifidobacterium*	[65]
Bajaj JS	USA	2014	30	*L. rhamnosus* GG	Among cirrhotic patients with minimal hepatic encephalopathy, reduced *Enterobacterales* and increased Clostridiales Family XIV Incertae Sedis and *Lachnospiraceae* relative abundance, but no change in cognition	[64]
Nagino T	Japan	2018	60	*L. casei* Shirota	Consecutive intake of fermented soymilk (containing isoflavone), and *L. casei* Shirota decreased the levels of *Enterobacterales*	[63]
Arnbjerg CJ	Denmark	2018	45	*L. rhamnosus* GG	Decrease in intestinal inflammation, along with a reduction of *Enterobacterales* in the gut microbiome among human- immunodeficiency-virus-infected individuals	[62]
Dall LB	Denmark	2019	31	*L. rhamnosus GG*	No effect on the risk of colonization with extended spectrum β-lactamase (ESBL)-*Enterobacterales*	[61]
Ljungquist O	Sweden	2020	80	Vivomixx^® 1^	No support of Vivomixx^®^ as being superior to the placebo for intestinal decolonization in adult patients with chronic colonization of ESBL-producing *Enterobacterales*	[47]
Ramos-Ramos JC	Spain	2020	8	*B. bifidum* and *L. acidophilus* (Infloran^®^)	Three weeks of a combination of prebiotics and probiotics decreased the intestinal load of OXA-48-producing *Enterobacterales*	[49]
Wieërs G	Belgium	2021	120	Bactiol duo^® 2^	Colonization with AmpC-producing *Enterobacterales* declined after the probiotic intervention	[45]

^1^ contains 4 *Lactobacillus* strains (*L. paracasei* 24733, *L. acidophilus* 24735, *L. delbrueckii* subspecies *bulgaricus* 24734, and *L. plantarum* 24730), 3 *Bifidobacterium* strains (*B. brief* 24732, *B. longum* 24736, and *B. infantis* 24737), and *S. thermophilus* 24731; ^2^ contains *S. boulardii*, *L. acidophilus* NCFM, *L. paracasei* Lpc-37, *B. lactis* Bl-04, and *B. lactis* Bi-07.

**Table 3 antibiotics-10-01086-t003:** Three clinical trials of dietary supplementation with probiotics for the eradication of gut *Enterobacterales* carriage registered at ClinicalTrials.gov, as posted from July 2008 to July 2021.

ClinicalTrials.gov Identifier	Official Title	First Posted	Study Design/Case Number	Probiotic Strain	Location	Outcome Measures	Status
NCT 00722410	Safety and efficacy study of eradication of carbapenem- resistant *Klebsiella* *p**neumonia**e* from the gastrointestinal tract by probiotics	25 July 2008	Open-label, randomized/60	VSL#3^® 1^	Jerusalem, Israel	Negative stool culture for carbapenem-resistant *Klebsiella pneumoniae*	Not yet recruiting
NCT 03967301	Prevention and decolonization of multidrug-resistant bacteria with probiotics	30 May 2019	Double-blinded, randomized/228	Bioflora^® 2^	Buenos Aires, Argentina	Risk of colonization and/or infection by carbapenem-resistant *Enterobacterales*	Not yet recruiting
NCT 04431934	Open-label, randomized study to assess the efficacy of a probiotic or fecal microbiota transplantation (FMT) on the eradication of rectal multidrug-resistant Gram-negative bacilli (MDR-GNB) carriage (PROFTMDECOL)	16 June 2020	Open-label, randomized/437	Vivomixx^® 3^	Barcelona, Spain	Eradication of rectal multidrug-resistant Gram-negative bacilli carriage	Recruiting

^1^VSL#3^®^: 4 *Lactobacillus* strains (*L. acidophilus*, *L. plantarum*, *L. casei*, and the *L. delbrueckii* subspecies *bulgaricus*), 3 *Bifidobacterium* strains (*B. breve*, *B. longum*, and *B. infantis*), and the *S. salivarius* subspecies *thermophilus*; ^2^ Bioflora^®^: *L. casei*, *L. plantarum*, *S. faecalis*, and *B. brevis*; ^3^ Vivomixx^®^: 4 *Lactobacillus* strains (*L. paracasei* 24733, *L. acidophilus* 24735, *L. delbrueckii* ssp *bulgaricus* 24734, and *L. plantarum* 24730), 3 *Bifidobacterium* strains (*B. brief* 24732, *B. longum* 24736, and *B. infantis* 24737), and *S. thermophilus* 24731.

## Data Availability

Not applicable since this work did not report experimental data.

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
