# Peer review of "The Potential of Probiotics to Eradicate Gut Carriage of Pathogenic or Antimicrobial-Resistant Enterobacterales"

_antibiotics, 2021, doi:10.3390/antibiotics10091086_

Round 1

Reviewer 1 Report

The manuscript entitled: “The potential of probiotics to eradicate gut carriage of pathogenic or antimicrobial-resistant Enterobacterales” falls within the scope of the journal. This is a very interesting review, providing some clear examples on the use of probiotics to reduce pathogenic resistant Enterobacterales in humans and in animals. The authors were able to describe previous results in a succinct manner. Even though it is a review manuscript with loads of information, it was easy to follow and enjoyable to read. Please find some comments below.
The title states “antimicrobial resistant enterobacterales”, are all the Enterobactereales mentioned in the manuscript “antibiotic resistant”? Please make sure to state specifically which ones are resistant or not. It seemed unclear throughout the manuscript.
Line 63-70 seems out of place in the introduction, the data in cats and dogs could be included below, perhaps together with rabbits in a separate subsection (see comments below) as “domesticated animals”.
Line 128-132: Rabbits are not considered livestock in several countries including the US. As a suggestion, there are several studies in cattle, layers, broilers, swine that could be utilized as a better example, please include more details in this area.
Table 1: The titles for each column should be merged and center for improved clarity.
Line 262-264: unclear writing, please re-write.
Figure 1: The first part of the figure mentions antibiotic exposure, are probiotics recommended for use after antibiotic exposure? Or are the antimicrobial resistant bacteria already present in the gut and causing an infection? This figure is somewhat confusing, please elaborate in the figure legend if this is in humans or animals or both, and if this is after an infection in the gut with confirmed antibiotic resistant bacteria.
For the tables, perhaps a different orientation within the page would make the text easier to follow, especially Figure 1 and 2 seem to be very crowded.
No numbering, line: “In contrast to the beneficial effects in infants, for adults, probiotic supplements might decrease but fail to eradicate potentially pathogenic Enterobacterales in the gut.” Suggestion: In contrast to the beneficial effects in infants, for adults, probiotic supplements might decrease potentially pathogenic Enterobacterales but fail to completely eradicate them in the gut.

Author Response

Dear the editor and reviewer of Antibiotics,

Enclosed, please find our revised manuscript entitled "The potential of probiotics to eradicate gut carriage of pathogenic or antimicrobial-resistant Enterobacterales ". We have considered very carefully for the concerns raised by the reviewers and made responsive alterations in the revised manuscript. We hope that our changes have satisfactorily clarified the points raised.

In answering the reviewers’ concerns, we made point-by-point responses to the comments summarized below. The revisions made in the manuscript were marked by red color, rather than track changes, because of the differences in the line numbers of the revised parts in the versions of “clear” manuscript and “track change” manuscript. We prefer to show the revisions in the clear manuscript in order to clearly indicate the revised parts by red color in the response letter. We appreciate the reviewers and the editor for their careful reading and insightful comments, and make our best efforts to improve the manuscript by responding to their comments.

We look forward to hearing from you.

Best Regards,

Wen-Chien Ko, MD

Division of Infectious Diseases, Department of Internal Medicine, National Cheng Kung University Hospital, No. 138, Sheng Li Road, Tainan, 70403, Taiwan

TEL: 886-6-2353535, ext. 3596   FAX: 886-6-2752038

E-mail: winston3415@gmail.com

Reviewer 1:

The manuscript entitled: “The potential of probiotics to eradicate gut carriage of pathogenic or antimicrobial-resistant Enterobacterales” falls within the scope of the journal. This is a very interesting review, providing some clear examples on the use of probiotics to reduce pathogenic resistant Enterobacterales in humans and in animals. The authors were able to describe previous results in a succinct manner. Even though it is a review manuscript with loads of information, it was easy to follow and enjoyable to read. Please find some comments below. The title states “antimicrobial resistant enterobacterales”, are all the Enterobactereales mentioned in the manuscript “antibiotic resistant”? Please make sure to state specifically which ones are resistant or not. It seemed unclear throughout the manuscript.
Reply: Thank you for your critical comment. The texts about antimicrobial-resistant Enterobacterales were chiefly discussed at section 1.5 “Probiotic supplementation to decrease gut pathogenic or antimicrobial-resistant Enterobacterales colonization in adults”. We stated them specifically in this section about “pathogenic Enterobacterales or “antimicrobial-resistant Enterobacterales” in line 9, 12, 13, 17 of page 6.

Line 63-70 seems out of place in the introduction, the data in cats and dogs could be included below, perhaps together with rabbits in a separate subsection (see comments below) as “domesticated animals”.

Reply: The texts about cats, dogs, and rabbits were moved to the section of 1.2, “Probiotic supplements to decrease gut carriage of Enterobacterales in livestock or domesticated animals” as suggested in line 38 of page 3, and line 45 of page 4 ~ line 2 of page 5.

Line 128-132: Rabbits are not considered livestock in several countries including the US. As a suggestion, there are several studies in cattle, layers, broilers, swine that could be utilized as a better example, please include more details in this area.

Reply: The paragraph about cats, dogs, and rabbits were moved to line 45 of page 4 ~ line 2 of page 5.

Table 1: The titles for each column should be merged and center for improved clarity.

Reply: Patient population and number were merged to improve clarify in Table 1.

Line 262-264: unclear writing, please re-write.

Reply: The sentence was revised as “but probiotics or synbiotics might be used in combination with a conventional bowel preparation to reduce the fecal carriage of Enterobacterales” in line 12-14 of page 7.

Figure 1: The first part of the figure mentions antibiotic exposure, are probiotics recommended for use after antibiotic exposure? Or are the antimicrobial resistant bacteria already present in the gut and causing an infection? This figure is somewhat confusing, please elaborate in the figure legend if this is in humans or animals or both, and if this is after an infection in the gut with confirmed antibiotic resistant bacteria.

Reply: The figure indicates that some clinical settings or medications promote the development of intestinal carriage of Enterobacterales in human, which may be asymptomatic but may later predispose to the occurrence of “primary” bloodstream infections. The legend of Figure 1 was modified as “In humans, some clinical settings or interventions might promote intestinal carriage of Enterobacterales. Probiotics might be beneficial in eradicating gut carriage of pathogenic or antimicrobial-resistant Enterobacterales.”.

For the tables, perhaps a different orientation within the page would make the text easier to follow, especially Figure 1 and 2 seem to be very crowded.

Reply: We agree the text editing made by the editorial office for Table 1 and table 2.

No numbering, line: “In contrast to the beneficial effects in infants, for adults, probiotic supplements might decrease but fail to eradicate potentially pathogenic Enterobacterales in the gut.” Suggestion: In contrast to the beneficial effects in infants, for adults, probiotic supplements might decrease potentially pathogenic Enterobacterales but fail to completely eradicate them in the gut.

Reply: The sentence in Abstract had been modified as suggested in line 30-31.

Reviewer 2 Report

The manuscript entitled: ‘The potential of probiotics to eradicate gut carriage of pathogenic or antimicrobial-resistant Enterobacterales’ raises the very important issue of modulating the carriage of multi-drug-resistant bacteria in the gastrointestinal tract. Probiotics are microorganisms known for a long time with pro-health effects, which are manifested, among others, by in antibacterial action.

For the above reasons, the manuscript appears to be valuable, however, in the opinion of the reviewer, it requires corrections before it is published.

Major revision:

  1. In section 1.1 provide the definition of probiotic according to international standards
  2. Add another section on the role of the microbiome in the host's organism and its formation and influence on the type of delivery, antibiotic therapy and infant formula feeding in the context of colonization by Enterobacterales
  3. Add an additional section on the safety of probiotics with an emphasis on possible infections such as sepsis, bacteremia and infective endocarditis caused by bacteria from supplemented probiotics. In addition, discuss the issue of possible transfer of genes encoding antibiotic resistance by probiotic bacteria.

Yours sincerely,

Reviewer

Author Response

Dear the editor and reviewer of Antibiotics,

Enclosed, please find our revised manuscript entitled "The potential of probiotics to eradicate gut carriage of pathogenic or antimicrobial-resistant Enterobacterales ". We have considered very carefully for the concerns raised by the reviewers and made responsive alterations in the revised manuscript. We hope that our changes have satisfactorily clarified the points raised.

In answering the reviewers’ concerns, we made point-by-point responses to the comments summarized below. The revisions made in the manuscript were marked by red color, rather than track changes, because of the differences in the line numbers of the revised parts in the versions of “clear” manuscript and “track change” manuscript. We prefer to show the revisions in the clear manuscript in order to clearly indicate the revised parts by red color in the response letter. We appreciate the reviewers and the editor for their careful reading and insightful comments, and make our best efforts to improve the manuscript by responding to their comments.

We look forward to hearing from you.

Best Regards,

Wen-Chien Ko, MD

Division of Infectious Diseases, Department of Internal Medicine, National Cheng Kung University Hospital, No. 138, Sheng Li Road, Tainan, 70403, Taiwan

TEL: 886-6-2353535, ext. 3596   FAX: 886-6-2752038

E-mail: winston3415@gmail.com

Reviewer 2:

The manuscript entitled: ‘The potential of probiotics to eradicate gut carriage of pathogenic or antimicrobial-resistant Enterobacterales’ raises the very important issue of modulating the carriage of multi-drug-resistant bacteria in the gastrointestinal tract. Probiotics are microorganisms known for a long time with pro-health effects, which are manifested, among others, by in antibacterial action.

For the above reasons, the manuscript appears to be valuable, however, in the opinion of the reviewer, it requires corrections before it is published.

Major revision:

  1. In section 1.1 provide the definition of probiotic according to international standards

Reply: “Probiotics, by definition, are live microorganisms and should remain viable when they reach the intended site of action which is typically the cecum and/or colon” had been added in line 42-43 of page 2.

  1. Add another section on the role of the microbiome in the host's organism and its formation and influence on the type of delivery, antibiotic therapy and infant formula feeding in the context of colonization by Enterobacterales

Reply: The role of the microbiome in the host's organism and the type of delivery, antibiotic therapy and infant formula feeding in the context of colonization by Enterobacterales had been illustrated in the first paragraph of Introduction, i.e., line 42 of page 1 ~ line 17 of page 2.

  1. Add an additional section on the safety of probiotics with an emphasis on possible infections such as sepsis, bacteremia and infective endocarditis caused by bacteria from supplemented probiotics. In addition, discuss the issue of possible transfer of genes encoding antibiotic resistance by probiotic bacteria.

Reply: An additional section on the safety of probiotics with an emphasis on possible adverse events has been added in line 7-19 of page 11.

Round 2

Reviewer 2 Report

In my opinion, the manuscript can be published.